# Involvement of Reactive Oxygen Species in Prostate Cancer and Its Disparity in African Descendants

**DOI:** 10.3390/ijms25126665

**Published:** 2024-06-17

**Authors:** Geou-Yarh Liou, Reauxqkwuanzyiia C’lay-Pettis, Sravankumar Kavuri

**Affiliations:** 1Center for Cancer Research and Therapeutic Development, Clark Atlanta University, Atlanta, GA 30314, USA; 2Department of Biological Sciences, Clark Atlanta University, Atlanta, GA 30314, USA; 3Department of Pathology, Augusta University Health, Augusta, GA 30912, USA

**Keywords:** reactive oxygen species, oxidative stress, inflammation, prostate cancer, racial disparity, African Americans

## Abstract

Reactive oxygen species (ROS) participate in almost all disorders, including cancer. Many factors, including aging, a high-fat diet, a stressful lifestyle, smoking, infection, genetic mutations, etc., lead to elevated levels of ROS. Prostate cancer, the most prevalent type of cancer in senior American men and the second leading cause of cancer mortality in American men, results from chronic oxidative stress. The doubled incident rate as well as the doubled mortality numbers of prostate cancer have persisted in African Americans in comparison with Caucasian Americans and other racial groups, indicating a prostate cancer disparity in African American men. In this review, we mainly focus on the latest findings on ROS in prostate cancer development and progression within the last five years to update our understanding in this area, as several comprehensive literature reviews addressing oxidative stress and/or inflammation in prostate cancer before 2020 are available. In addition to other known factors such as socioeconomic disadvantage, cultural mistrust of the health care system, etc. that are long-existing in the African American group, we also summarize the latest evidence that demonstrated high systemic oxidative stress and inflammation in African Americans for their potential contribution to the racial prostate cancer disparity in this population.

## 1. Introduction

Reactive oxygen species (ROS) are highly reactive molecules that contain oxygen. ROS are characterized by unpaired and unstable electrons that readily oxidize other molecules. ROS includes superoxide anion, hydrogen peroxide, nitric oxide, peroxynitrite, hydroxyl radical, xanthine oxidase, and singlet oxygen. Inflammation is a process that produces ROS. Elevated ROS levels are commonly referred to as inflammatory or pro-inflammatory. Major sources of ROS inside cells include mitochondrial respiration and oxidative metabolism, NADPH oxidases, the cellular response to xenobiotics and cytokines, peroxisomes, cytochrome p450 [1], and various enzymes. Production of ROS is not limited to cell types or any cellular locations. In addition to immune defense against pathogens during infection [2], under normal physiological conditions, ROS are involved in many essential cellular processes, including cell signaling, homeostasis, cell growth, proliferation and differentiation, and cell survival. During these processes, only a low concentration of ROS is required for executing the described tasks [3]. To maintain a low concentration of intracellular ROS, a balance between ROS production and the generation of antioxidants, molecules that break down ROS, is essential. 

When excessive ROS levels are present inside a cell, these highly reactive molecules result in non-specific oxidation of nucleic acids, lipids, and other macromolecules, including DNA adducts, lipid peroxidation, and protein carbonylation [4,5]. All of these damaged biomolecules, when existing in large amounts without being efficiently repaired or eliminated in time, eventually lead to cell death [6]. In addition, the damaged biomolecules, for example, 8-oxodG, which is the most commonly formed DNA adduct under high levels of ROS, have been shown to enhance carcinogenesis [7,8,9]. 

## 2. Sources of Intracellular ROS Production and Cell Systems That Detoxify ROS

There are several enzymes that participate in producing ROS in cells. To maintain the intracellular ROS levels within a constant range, cells also have systems to decrease the generated ROS and subsequent free radicals by neutralizing them through antioxidants. 

### 2.1. Enzymes and Organelle That Produce ROS

#### 2.1.1. NADPH Oxidases

One of the sources for intracellular ROS is NADPH oxidase (NOX). The NOX family consists of seven enzymes, five of which are NOX subunits, including NOX1–5, and the other two are DUOX subunits, DUOX1 and DUOX2 [10]. In addition to generating ROS, NADPH oxidases are also involved in homeostatic processes such as cell proliferation, immune response, maintaining normal cellular functions, and regulating organ functions [10]. Overproduction of ROS in the brain, which results in neuron cell death, has been associated with neurodegenerative diseases including Alzheimer and Parkinson disease [11]. It has been shown that NOX2 is the major NOX isoform in the brain, and overexpression of NOX2 led to oxidative stress, which caused protein misfolding and aggregation in neurons and eventually resulted in cell death of neurons [12,13,14]. In addition, NOX1-mediated ROS has also been shown to degrade dopaminergic neurons in animal models for Parkinson’s disease [15,16]. In cancer, ROS production can lead to mutations and crosslinking of DNA. NOX4 is expressed in various tumors and melanomas, and NOX5 is expressed in gastric and prostate cancers, as well as other cancers [10,17,18]. In addition, NOX1, NOX2, NOX4, and NOX5 are all reported to regulate angiogenesis, an essential process in cancer metastasis [19,20,21]. 

The DUOX1 and DUOX2 NADPH oxidases are expressed primarily in the thyroid, with the major function of DUOX2 being iodine synthesis [22]. DUOX dysregulation is associated with hypothyroidism and goiter. Increased DUOX2 expression can lead to decreased thyroperoxidase (TPO) expression, leading to increased oxidative stress and DUOX2 dysfunction [23]. 

#### 2.1.2. Mitochondria 

Mitochondria, known as cellular powerhouses, produce ATP through oxidative phosphorylation. The electron transport chain is a series of transmembrane protein complexes that pass through electrons between the electron carrier molecules, such as NADH and FADH2, during oxidative phosphorylation. The protein complexes for this energy transport system are Complexes I, II, III, IV, and V, and Complexes I and II are the primary producers of ROS. 

#### 2.1.3. Cytochrome P450 

There are many cytochrome enzymes that are found across all domains of life, as they are evolutionarily conserved. The Cytochrome P450 family of enzymes, known as CYP450, are involved in the electron transport chain, cell-signaling processes, and detoxification of xenobiotics. CYP450 produces ROS as a product of detoxification and releases them into the cytosol compartment of the cell. CYP450 is typically found in the liver and can contribute to oxidative stress as well as liver diseases. When alcohol is consumed, CYP450 metabolizes alcohol and generates ROS as a byproduct during this process [24,25]. Of note, these byproduct ROS can lead to lipid peroxidation, followed by DNA modifications. 

#### 2.1.4. Xanthine Oxidases 

Xanthine oxidase is involved in purine catabolism across multiple species. In humans, hypoxanthine is converted to xanthine, which is then converted to uric acid. Another alternative name for xanthine oxidase is xanthine dehydrogenase, as it uses oxygen from water to complete the process [26]. This enzyme can be found in the extracellular compartments and peroxisomes, but is primarily located in the cytosol [27]. The ROS generated by xanthine oxidase include superoxide, hydroxyl radicals, and hydrogen peroxide. It has been reported by Kuppusamy and Zweier et al. that hydroxy radicals were increased when there was more hydrogen peroxide, whereas the levels of hydroxyl radicals were reduced in response to a superoxide enriched condition [28], thus suggesting an increase in hydroxyl radical production via xanthine oxidase in the presence of hydrogen peroxide. 

### 2.2. Antioxidants 

To maintain a redox balance in the cellular environment, it is essential to have an antioxidant system to counteract the produced ROS to keep the total cellular ROS levels within a steady range. Three major systems that function to detoxify high levels of ROS include superoxide dismutase (SOD), catalase (CAT), and glutathione peroxidase (GSH) [29]. SODs are responsible for ROS decomposition [30] by converting superoxide to hydrogen peroxide. Catalase acts similarly on ROS by breaking down hydrogen peroxide to become water. In addition to catalase, glutathione peroxidase can also reduce hydrogen peroxide to water. It has also been reported that GSH also reduced lipid hydroperoxides that led to DNA adducts 8-oxo-2′-deoxyguanosine to alcohols [6]. The FoxO family of transcription factors has been shown to regulate and promote antioxidant defense through post-translational modifications [31,32,33]. Because ROS is neutralized by antioxidants and higher levels of ROS do increase an individual’s chance of getting cancer, it is believed that the uptake of antioxidants would reduce or prevent cancer. However, several lines of evidence revealed the pro-tumor role of antioxidants [34,35].

## 3. ROS in Prostate Cancer Development, Progression, and Dissemination 

Prostate cancer is the most common type of cancer in senior men in the US and in more than half of the countries in the world [36]. In the United States, it is the second-leading cause of cancer death in men. The new case number of prostate cancer is on the rise, probably because of the increasing and aging population in the US. Multiple risk factors, including old age, family history, race, diet, consistently elevated testosterone levels, etc., may contribute to the susceptibility to prostate cancer [36,37]. The disease tends to be more aggressive in men who are diagnosed with prostate cancer at a younger age as compared to those who are older [36]. In addition, genetic mutations, including BRCA1 and BRCA2, have also been reportedly associated with prostate cancer [37]. Ethnicity is another potential risk for prostate cancer in senior men. It has been shown that the incidence and mortality of prostate cancer are doubled in the African American group among all other racial groups, whereas Asian American men have the lowest chance for prostate cancer [38,39,40], thus indicating prostate cancer disparity. 

Accumulating evidence also indicated certain dietary factors, such as a high-fat diet and processed meats, are tightly linked to prostate cancer onset and progression and have been recently summarized in an updated review by Oczkowski et al. [41]. Interestingly, dietary components that are involved in antioxidants have been reported to restrain prostate tumor cells from progressing to the next stage. It has been shown that treating cultured human prostate cancer cell lines, including PC-3 and LNCaP, with sinapic acid, a derivative of hydroxycinnamic acid that is commonly found in various vegetables and fruits, induced apoptosis by upregulating caspase 3, caspase 7, and Bax in these cells [42]. Furthermore, it also decreased the invasiveness of prostate cancer cells by reducing MMP9 expression. On the other hand, Vivarelli et al. recently showed that vitamin E increased gene expression of cytochrome P-450 enzymes, including CYP1A1, CYP1A4, CYP4F2, CYP2C9, and CYP2B6, in human non-tumorigenic prostate epithelial RWPE-1 cells [43]. All these cytochrome P-450 enzymes were associated with pro-carcinogens such as polychlorinated biphenyls, resulting in elevated levels of intracellular ROS and upregulation of inflammatory genes such as cyclooxygenase-2 (Cox-2) in RWPE-1 cells [41]. Furthermore, they also showed that when giving vitamin E daily to rats through intraperitoneally injection for 1–2 weeks, the increase of the CYP enzyme upregulation and activities as well as the subsequent inflammation were detected in the prostate of rats, thus implicating vitamin E as a co-carcinogen in the prostate.

Accumulating evidence has demonstrated a link between ROS and prostate cancer. Kumar et al. detected higher levels of intracellular ROS, including H_2_O_2_ and superoxide, in cultured human prostate cancer cells, including LNCaP, Du145, and PC3, than in prostate primary epithelial cells, as well as in the cultured prostate epithelial cell line RWPE-1 [44]. In addition, inhibition of ROS production from the NAPDH oxidase (Nox) system by treating prostate cancer cell lines with the Nox inhibitor diphenyliodonium impaired cell migration ability, anchorage-independent colony formation, and cell proliferation of prostate cancer cells, suggesting an essential role of ROS in malignant cell behaviors of prostate cancer. Lim et al. showed increased levels of hydrogen peroxide and Nox1 expression in human prostate cancer tissue samples [45]. Höll et al. demonstrated increased mRNA levels of Nox1, Nox2, and Nox5 in the majority of cultured human malignant prostate cancer cell lines, including LNCaP, VCaP, PC-3, and Du145 cells, when compared to benign prostate epithelial cells [46]. Furthermore, knockdown of Nox5 via short hairpin Nox5 (shNox5) in PC-3 cells impaired cell proliferation of PC-3 cells, suggesting ROS contributed to prostate cancer cell growth. The same effect of shNox5 was observed in all other malignant prostate cancer cell lines. 

C-creative protein (CRP), a protein produced from the liver, is increased in response to inflammation and is commonly detected in the serum sample in clinic; therefore, a high CRP number indicates elevated inflammation in the body. In a study of a total of 261 prostate cancer patients who were treated with radiotherapy, it showed that increased levels of plasma CRP (≥8.6 mg/L) are negatively correlated with poor cancer-specific survival, overall survival, as well as disease-free survival, indicating it is a prognostic factor in prostate cancer patients (see Table 1 for summarized results) [47]. In addition, this correlation is independent of tumor stage, Gleason scores, and prostate-specific antigen (PSA) levels. Recently, in another clinic study consisting of 524 patients to evaluate the correlation between the CRP and testosterone levels and the risk of clinically significant prostate cancer (Gleason score ≥ 7) at the time of prostate biopsy, Gómez-Gómez et al. showed that higher CRP levels (>2.5 mg/L) were positively associated with a higher risk of clinically significant prostate cancer [48]. Furthermore, higher circulating CRP levels were positively linked to a higher Gleason score in prostate cancer patients. However, there was no correlation between testosterone levels and the risk of clinically significant prostate cancer in this study. The latest study from Beyaztas et al. showed that levels of oxidative stress and inflammation were significantly elevated in groups of patients with benign prostatic hyperplasia (BPH) and prostate cancer with different risk levels as compared to the healthy individuals’ group [49]. These levels were measured based on the oxidative stress index, categorizing the total level of oxidants and antioxidants, and the serum levels of several cytokines, including IL-10, IL-1β, IL-6, and TNF. The levels of total antioxidant status were lower in all prostate cancer patient groups. Furthermore, patients who received surgeries had decreased levels of oxidative stress and inflammation in their bodies after surgical operations. Shukla et al. also reported that serum PSA levels and levels of 8-OHdG, an oxidized nucleoside of DNA, were higher in high-risk prostate cancer patients than in age-matched healthy individuals, whereas higher levels of glutathione S-transferase (GST), an enzyme to catalyze the conjugation of GSH with abundant hydrophobic and electrophilic intermediates, were present in the healthy control group [50]. However, there was no difference in plasma levels of catalase, glutathione peroxidase, glutathione reductase, superoxide dismutase, or lipid peroxide formation between the high-risk prostate cancer group and the age-matched healthy control group [50]. Overall, these reports demonstrated that oxidative stress led to prostate cancer development, progression, and invasiveness.

## 4. Prostate Cancer Disparity in African Americans

Among different racial groups, accumulating evidence has demonstrated that African Americans and/or men with African ancestry are affected the most by prostate cancer disparity, as they have the highest incidence rate and cancer deaths from prostate cancer [40,51]. Many factors can contribute to the prostate cancer disparity in African American men, and these include socioeconomic status, biological factors, cultural mistrust of the health care system [52], poor communication between African American men and their physicians [53,54], and a lack of knowledge on prostate cancer in black families and communities [55,56], etc. Among them, the most well-described factors are socioeconomic status and certain biological factors. 

### 4.1. Socioeconomic Factors

Socioeconomic status, known as SES, is a measure that places a group or individual into different levels, typically high, middle, or low, according to their education, income, and occupation. The SES is positively associated with access to financial, educational, social, and health resources. The African American group overall has a lower SES among all other racial groups, thus revealing the disadvantage of this racial group in the accessibility to prostate cancer screening, diagnosis, and treatment. Many published reviews have been primarily focused on summarizing the findings linking the SES of African Americans with low grade prostate cancer to the high incidences and mortality of African American men [57,58,59,60]. In this review, we focus solely on the latest updates and some very crucial findings in this field. 

Lower levels of SES are often thought to be associated with prostate cancer. In a study with a cohort of 98,484 incident prostate cancer cases in total among African Americans, non-Hispanic Whites, Hispanics, and Asian/Pacific Islanders in California [61], it was demonstrated that higher levels of SES, regardless of race, were positively associated with an increased risk of prostate cancer. This is probably because people with higher levels of SES have not only significantly higher stress related to jobs and finances but also a lack of physical activity. For prostate cancer deaths, higher levels of SES were associated with a lower mortality rate of prostate cancer fatalities [61]. Furthermore, African American men had a 2–5-fold increased risk of prostate cancer deaths as compared to Caucasian Americans across all levels of SES, thus suggesting other factors also account for the prostate cancer disparity in African Americans. 

SES levels also reflect the security of food and housing, which is critical to prostate cancer onset and outcome. Pichardo et al. used the neighborhood deprivation index to link prostate cancer and the immune systems of African and Caucasian American residents in the Baltimore area [62]. This index is based upon factors including the percentage of households in pauperism, the percentage of female-headed households with dependent children, the percentage of households acquiring public assistance, the percentage of households with annual income less than USD 30,000, the percentage of males and females out of work, and the percentage of manager occupation [62]. In this cohort study of 405 African American men and 364 Caucasian American men for prostate cancer cases and age-/race-matched population controls (479 African Americans and 544 European Americans), they showed that the neighborhood deprivation scores were higher in the African American group as compared to the Caucasian American group regardless of controls or cases of prostate cancer. In addition, when combining both racial groups, neighborhood deprivation promoted a 65% elevated risk of prostate cancer without taking SES into account, indicating the risk of prostate cancer is independent of SES. Furthermore, neighborhood deprivation was positively associated with metastatic prostate cancer among all men, regardless of their race or SES [62]. Most importantly, they also showed that biological pathways related to inflammation and tumor immunity suppression were significantly impacted by SES, and only chemotaxis pathways were solely associated with neighborhood deprivation instead of individual SES. This suggests that neighborhood socioeconomic deprivation can influence the systemic immune function and inflammation of individuals, which links socioeconomic status to biological factors, for example, oxidative stress levels, inflammation/immunity, etc. The dietary inflammatory index (DII) is a scoring system to evaluate the inflammatory potential of foods and nutrients in an individual diet. Using this index as a surrogate for SES levels, Vidal et al. reported that among 254 prostate cancer cases with 328 controls from black and white US veteran men, a pro-inflammatory diet with a higher DII score was positively associated with black American men [63]. In addition, it was also positively correlated with high-grade prostate cancer. Altogether, it indicates the impact of SES through diet and food security on prostate cancer disparity in black men. 

This risk for prostate cancer mortality in all African American men is unequal, according to a recent report that was set to investigate prostate cancer survival in African American men in different regions across the US through 17 geographic registries within the Surveillance, Epidemiology, and End Results (SEER) databases [64]. The geographic areas included in this cohort study were the regions of Atlanta, Georgia; Detroit, Michigan; Greater Georgia; Rural Georgia; San Francisco-Oakland metropolitan area, California; San Jose-Monterey, California; Louisiana; New Jersey; New Mexico; Hawaii; Iowa; Kentucky; Los Angeles, California; Utah; and Seattle. When analyzing according to patients’ ethics and Gleason scores, results showed that African American men in the areas of Atlanta, Georgia; Greater Georgia; Louisiana; and New Jersey showed worse prostate cancer-specific survival in both Gleason grade group 1 and grade groups 2–5. It also revealed the greatest race-based prostate cancer survival difference among African American men with low-grade prostate cancer, especially in the Atlanta, Georgia area, where there was a 5-fold increase in prostate cancer mortality risk compared to other metropolitan areas in the US. 

### 4.2. Biological Factors

Whether there are unique biological factors that lead to prostate cancer disparity in the African American group has been under debate until a very recent report from a cohort study carried out in the veteran affairs (VA) healthcare system that allows all access to qualified veterans regardless of their race [65]. This study included 7,889,984 veterans undergoing routine care in VA hospitals nationwide from 2005 through 2019, and the result revealed that African American men were at least 2 years younger when diagnosed with localized prostate cancer and had higher PSA levels in comparison with white men. In addition, African American veterans had around a 2-fold higher incidence of localized and metastatic prostate cancer than white veterans. Among veterans screened for prostate cancer, African American men had a 29% increased risk of prostate cancer detection on a diagnostic prostate cancer biopsy compared to white men. Altogether, it indicated unknown biological factors in African American men resulting in the high incidence and higher mortalities of prostate cancer in this population as compared to other racial groups. 

The biological factors associated with African Americans leading to their susceptibility to prostate cancer may arise from their lifestyle, cultural influence, and/or evolution. Obesity is more prevalent in African American communities in the US than in other racial groups [66]. It has been reported that obese African American men are at a high risk for prostate cancer, with a 122% increased risk for low-grade prostate cancer and an 88% increased risk for high-grade prostate cancer as compared to those who have a normal weight [67]. In a co-culture system of the cultured human prostate cancer cell lines, including PC-3, Du145, and C4-2B cells, with murine differentiated mature adipocytes, it has been demonstrated that cancer cells in the tumor-invasive front can promote the release of free fatty acids (FFAs) from the lipolysis of adipocytes [68]. In addition, cancer cells then elevated ROS via Nox5 by utilizing these free fatty acids around them to potentiate their own invasiveness and migration ability to spread to other parts of the body via upregulation of HIF1 α and MMP14. Most importantly, this signaling axis of FFAs/Nox5/ROS/HIF1α/MMP4 leading to invasiveness of prostate cancer cells is enhanced in obese men that have more FFAs to start with, thus implicating a link between African American men and aggressive prostate cancer. 

## 5. ROS and Inflammation Contribute to Prostate Cancer Disparity in African Americans

Chronic inflammation, which is associated with persistently low-elevated levels of intracellular ROS, can result from numerous factors, including the environment, obesity, a high-fat diet, aging, high stress, genes, etc. Using systemic inflammatory markers, including highly sensitive serum CRP (with a detection limit of 0.12 mg/L) and white blood cell count from patient serum samples, a large-scale study containing 509 prostate cancer patients and 6761 men without prostate cancer that was recently reported from Norway showed a systemic inflammatory score that utilizes a combination of highly sensitive serum CRP levels and white blood counts for evaluating the prostate cancer risk in men. Those with a higher systemic inflammatory score had a 28% increased risk for prostate cancer, especially being diagnosed with metastatic prostate cancer, in comparison with those whose systemic inflammatory score is low [69]. 

Several lines of evidence have revealed that African Americans possess a more inflammatory and oxidative system in the body as compared to Caucasian Americans. Deo et al. reported that in a cohort study containing 18 African American men and 16 Caucasian American men with similar age (20–24 years old), weight, BMI, and blood pressure, significantly higher levels of plasma protein carbonylation and intracellular superoxide production were found in peripheral blood mononuclear cells of the African American group [70]. Feairheller et al. also reported that in two racial groups of African and Caucasian Americans with similar age and comparable body mass index and blood pressure, when using NO levels, SOD activity, IL-6 levels, total antioxidant capacity, and protein carbonyl content in plasma samples to indicate overall oxidative stress level and inflammation in the body, higher levels of protein carbonylation, SOD activity, and total antioxidants were present in the African American group than the Caucasian American group [71]. Furthermore, the same results were also found in human umbilical vein endothelial cells (HUVECs) derived from these two racial groups in in vitro models. Of note, the expression of p47phox, Nox2, and Nox4 proteins, all of which are involved in forming NADPH oxidase that produces intracellular ROS, was significantly increased in the HUVEC cells derived from African Americans [71]. Altogether, it suggests a racial difference, especially for African Americans, in oxidative stress and inflammation. 

While using the neutrophil-to-lymphocyte ratio (NLR) and monocyte-to-lymphocyte ratio (MLR) from the peripheral blood samples as two indicators for the systematic inflammation to evaluate the systematic inflammation in prostate cancer susceptibility in black and white men, Rundle et al. showed that in the black men group, prostate cancer cases had a higher MLR trend as compared to the controls; however, the MLR is significantly lower in prostate cancer cases in the white men group than the controls [72]. Furthermore, a higher MLR was present in prostate cancer cases (either aggressive or non-aggressive) in the black men group than in the white men group, suggesting higher levels of systemic inflammation were present in black men than in the white men group. In addition, it also indicated a distinct inflammatory profile existed between two different races in the context of prostate cancer development. 

In a microarray study that was to identify specific genes associated with prostate cancer in African Americans than in Caucasian American men using the prostate cancer tissue samples, Singhal et al. reported an upregulation of genes involved in inflammation, TNF signaling, apoptosis, androgen response and epithelial-mesenchymal-transition (ENT) in African American men than in Caucasian American men [73]. Another study from Awasthi et al. that utilized a similar method to identify the immune-specific genes enriched in African American (AA) prostate cancer than in European American (EA) prostate cancer has revealed six genes, including IFTM3, IFI6, ANPEP, CD38, MT2A, and IFI44L [74]. Among them, IFTM3 (IFN-inducible transmembrane protein 3), a proinflammatory marker, was positively associated with the increased risk of recurrence of prostate cancer, specifically in the AA group. In addition, results also showed that AA had higher IFNα and IFNɣ expression and a low overall DNA repair genomic score as compared to the EA group [74]. Similarly, in South Africa, a country with diverse racial populations including black South-Eastern Bantu-speakers/colored, whites of European origin and a population group from the Indian subcontinent, cancer researchers who investigated the relationship between the inflammatory metabolic profile and prostate cancer in racial difference [75] reported that extremely high levels of inflammation markers such as glycoprotein acetylation (ClycA) and acetylneuraminic acid (GlycB), along with a very low level of histidine in the plasma samples, were present in the metabotype IV group that has a unique signature and is almost exclusively formed by patients who are at a very high risk and by solely metastatic prostate cancer patient. In addition, it also demonstrated reduced levels of all VLDLs (very low-density lipoproteins), including VLDL1-5, in the metabotype III group (with an average PSA of 25 ng/mL), as compared to the metabotype II group (with an average PSA of 18 ng/mL), thus adding another new marker to prognose and monitor prostate cancer progression in patients who are African descendants [75].

Inflammation and oxidative stress can result from the cells surrounding prostate tumors, such as tumor-infiltrating immune cells and prostate cancer-associated fibroblasts. In a recent study, Gillard et al. characterized the isolated prostate fibroblasts from the prostate cancer tissues of AA and EA regarding their function in prostate cancer progression [76] (see Figure 1 for summarized findings). They reported that prostate cancer-derived fibroblasts from AA prostate cancer tissues (PrF-AA) had increased growth in response to androgens, FGF2, and PDGF. In addition, the conditioned media from PrF-AA significantly elevated the cell proliferation and cell migration abilities of all cultured human prostate cancer cell lines, including LNCaP, C4-2B, PC-3, etc., in vitro [76]. Furthermore, results from the transcriptome analysis of both PrFs via RNA-Seq revealed that PrF-AA had high levels of secreted factors such as BDNF, CHI3L1, DPPIV, FGF7, IL18BP, IL-6, and VEGF, suggesting that inflammatory factors secreted from PrF-AA accelerated prostate cancer growth and potential invasiveness in AA men. Maynard et al. found that CD66ce^+^ neutrophils were increased in prostate cancer as compared to benign areas in low-grade prostate cancer as compared to high-grade [77]. Neutrophils were also seen in cancer regions of white men compared to those in black men [77] based on the use of CD66ce as a neutrophil marker, CD68 as a pan-macrophage marker, CD80 as a M1 inflammatory macrophage marker, and CD163 as a M2 immunosuppressive macrophage marker to detect these tumor-infiltrating immune cells in human prostate tissues. The tested tissues included normal prostate, benign, and cancer tissues from black and white men groups. In addition, the numbers of both M1 and M2 macrophages were higher in the cancer as compared to the prostate benign tissues, and the increased numbers of CD163^+^ macrophages detected in higher-grade cancers were higher than in low-grade cancers. However, more CD80^+^ macrophages were present in the benign tissues of black men than those of white men [77]. The distribution of these infiltrating neutrophils and macrophages in the surrounding areas of prostate epithelial cells may implicate an inflammatory tumor microenvironment to expedite prostate tumor development in AA men, especially at the low-grade tumor stage. 

Hormone therapy, known as androgen deprivation therapy (ADT), is used to reduce levels of androgen that fuels cancer cell growth in prostate cancer patients who are in the early stage of the disease. Regarding whether androgen deprivation therapy is more beneficial to men with African ancestry, Bassey et al. evaluated the levels of oxidative stress and antioxidant status in Nigerian prostate cancer patients who were under androgen deprivation therapy or non-treatment/naïve and in Nigerian healthy men [78]. By using serum levels of malondialdehyde (MDA), nitric oxide (NO), total plasma peroxide (TPP), and total antioxidant capacity (TAC) as indicators for oxidative stress, they showed that prostate cancer patients had a higher PSA, waist-hip ratio, NO, TPP, oxidative stress index (OSI), and MDA than the healthy individual group. Furthermore, the patients who underwent androgen deprivation therapy had significantly lower PSA, NO, and all other comparable levels of MDA, TAC, and TPP than those without treatment [78]. Of note, the levels of serum MDA were positively correlated with the duration of ADT, suggesting that ADT, although decreasing serum PSA levels in Nigerian prostate cancer patients, increased oxidative stress, which may be the potential cause of developing castration-resistant prostate cancer in patients. 

One of the main sources of oxidative stress is mitochondria. Using the matched blood and prostate cancer tissue samples from a group of 87 South African men—77 men with African ancestry and 10 Caucasian men—for deep sequencing of complete mitochondrial genomes, results showed there were 144 unique somatic mitochondrial DNA (mtDNA) single nucleotide variants (SNVs) present in African men as compared to Caucasian men [79]. In addition, 80 of the somatic mtDNA SNVs were present in 39 men with African ancestry with aggressive prostate cancer. The number and frequency of somatic mtDNA SNVs were positively correlated with serum PSA levels and prostate cancer stage in these men. Altogether, it suggested the polymorphism in these somatic mtDNAs resulted in a high incidence and aggressiveness of prostate cancer in African descendants. 

## 6. Conclusions

High oxidative stress, known as elevated levels of ROS or increased inflammation, promotes the initiation, progression, and dissemination of prostate cancer. An elevated systemic inflammation has been detected in the AA group, which has the highest incidence and mortality of prostate cancer among all other racial groups. The cause of this racial disparity in AA men is linked not only to external factors such as socioeconomic status, neighborhood deprivation, cultural mistrust, etc., but also to biological factors including higher oxidative stress from more infiltrating immune cells, more inflammatory metabolites, etc. Efforts from all aspects are required to bring down the chronic inflammation/oxidative stress in this population to reduce the prostate cancer incidence or to enhance the oxidative stress threshold to trigger cancer cells to undergo apoptosis as the therapeutic strategy to decrease prostate cancer fatality in AA men.

## Figures and Tables

**Figure 1 ijms-25-06665-f001:**
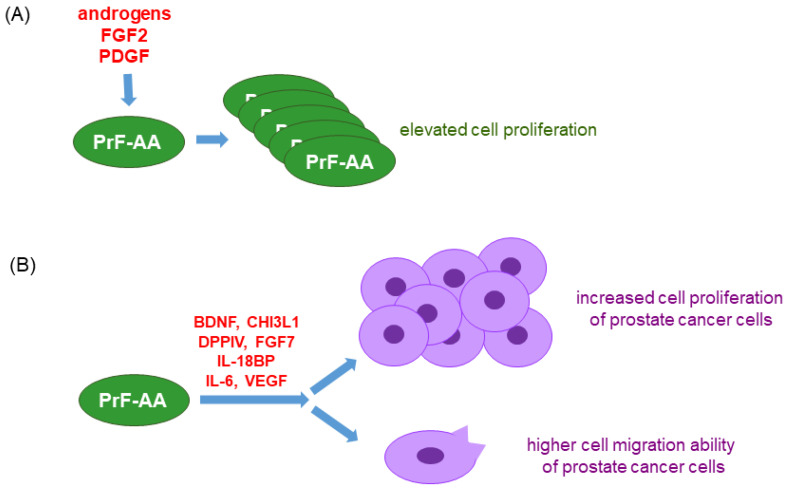
Summary of the effects of prostate cancer-associated fibroblasts derived from African American tissue samples (PrF-AA) as compared to those derived from European American tissue samples. (**A**) When treating both PrF-AA and PrF-EA cells with androgens, such as FGF2 or PDGF, PrF-AA cells had higher cell growth than PrF-EA cells. (**B**) The conditioned media of PrF-AA increased cell proliferation and cell migration of the cultured prostate cancer cell lines, possibly via their secreted factors as indicated. Abbreviation: FGF: fibroblast growth factor; PDGF: platelet-derived growth factor; BDNF: brain-derived neurotrophic factor; CHI3L1: chintinase-3-like protein 1; DPPIV: dipeptidyl peptidase IV; IL-18BP: IL-18 binding protein; VEGF: vascular endothelial growth factor.

**Table 1 ijms-25-06665-t001:** Summary of recent findings on oxidative stress in prostate cancer in clinic studies.

Oxidative Stress Status	Oxidative Stress Indicator	Patient Treatment	Patient Survival	Tumor Stage	Gleason Score/PSA Levels	Reference
high	CRP ≥ 8.6 mg/L	radiotherapy	low	no correlation	no correlation	[45]
high	CRP > 2.5 mg/L	N/A	ND	advanced	high	[46]
high	Total level of oxidants and antioxidants, IL-10, IL-1β, IL-6 and TNF levels	Patients who received surgeries had reduced oxidative stress levels	ND	ND	ND	[47]
high	8-OHdG, GST *	N/A	ND	ND	high	[48]

N/A: not applicable. ND: not determined. *: lower glutathione S-transferase (GST) levels in prostate cancer patients as compared to the healthy individual group.

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
