# Peer review of "Involvement of Reactive Oxygen Species in Prostate Cancer and Its Disparity in African Descendants"

_ijms, 2024, doi:10.3390/ijms25126665_

Round 1

Reviewer 1 Report

Comments and Suggestions for Authors

This review article by Liou et al. describes involvement of reactive oxygen species in prostate cancer and its disparity in African descendants. This is an interesting and valuable in application of molecular epidemiology, oncology and public health. However, some points need clarifying and certain statements require further justification.  These are given below.

Major points:

1)   This paper appears to be a NARRATIVE REVIEW focused on recent literature, but it is not sufficiently documented and narrowed down to include the most important 5 chapter.

2)   The theme of this is challenging, but the relationship between the biological phenomenon of ROS and socioeconomic factors is not adequately described although difficult.

Minor points:

1)   If there is a paper or conference presentation by the authors, it should be cited. The abrupt insertion of Figure 1 or Figure 2 is valuable information, but is outside the rules.

2)   Figure 3 is too simplistic. Although ROS has been implicated in both prostate cancer and prostate enlargement, they cannot be placed on the same line in the carcinogenesis sequence.

Author Response

Please see the attachment for this point-by-point response. Thank you.

Reviewer 2 Report

Comments and Suggestions for Authors

It is a timely review addressing differences of ROS and associated biomarkers between prostate cancer patients of African and European ancestries. However both over all presentation and data needs a careful edit.

Following needs to be addressed:

1. Manuscripts needs better organization  and thorough edit for the  presentation and grammar.

2. Key findings (Cytokines, CRP etc) can be presented in the form of table for better read.

3. Figure 1: How many specimens were analyzed for 4HNE  protein adducts in prostate cancer specimens and what was the frequency of this biomarker alteration for African American and Caucasian American Men. 

Comments on the Quality of English Language

 Manuscripts needs better organization  and thorough edit for the  presentation and grammar.

Author Response

(The authors gave the same response as above.)

Round 2

Reviewer 1 Report

Comments and Suggestions for Authors

This review article by Liou et al. describes involvement of reactive oxygen species in prostate cancer and its disparity in African descendants. Although this manuscript has been revised in accordance with the reviewers' suggestions, there is a major problem. In instructions for author submitting  to IJMS, “No new, unpublished data should be presented” indicates. Thus, Even if the authors wish, and even if the data are valuable, they cannot be published due to the definition of a NARRATIVE REVIEW. However, if the data has been published and described somewhere, it can be included in the form of a citation.

In addition, the relationship between the biological phenomenon of ROS and socioeconomic factors is important in this manuscript. The authors should search and speculate deeply enough and not only on prostate cancer.e.g. Oxid Med Cell Longev. 2021; 2021: 9965916.

Comments on the Quality of English Language

N/A

Author Response

Please see the attachment for the detailed point-to-point response. Thank you.

Round 3

Reviewer 1 Report

Comments and Suggestions for Authors

N/A